# Short-Term Joint Effects of PM_10_, NO_2_ and SO_2_ on Cardio-Respiratory Disease Hospital Admissions in Cape Town, South Africa

**DOI:** 10.3390/ijerph19010495

**Published:** 2022-01-03

**Authors:** Temitope Christina Adebayo-Ojo, Janine Wichmann, Oluwaseyi Olalekan Arowosegbe, Nicole Probst-Hensch, Christian Schindler, Nino Künzli

**Affiliations:** 1Department of Epidemiology and Public Health, Swiss Tropical and Public Health Institute, Basel, Kreuzstrasse 2, 4123 Allschwil, Switzerland; oluwaseyiolalekan.arowosegbe@swisstph.ch (O.O.A.); nicole.probst@swisstph.ch (N.P.-H.); christian.schindler@swisstph.ch (C.S.); nino.kuenzli@swisstph.ch (N.K.); 2Faculty of Medicine, University of Basel, 4056 Basel, Switzerland; 3Faculty of Health Sciences, School of Health Systems and Public Health, University of Pretoria, Pretoria 0002, South Africa; janine.wichmann@up.ac.za

**Keywords:** ambient air pollution, cardiovascular disease, respiratory disease, multi pollutant, short-term, DLNM, Cape Town, South Africa, time-series analysis

## Abstract

Background/Aim: In sub-Sahara Africa, few studies have investigated the short-term association between hospital admissions and ambient air pollution. Therefore, this study explored the association between multiple air pollutants and hospital admissions in Cape Town, South Africa. Methods: Generalized additive quasi-Poisson models were used within a distributed lag linear modelling framework to estimate the cumulative effects of PM_10_, NO_2_, and SO_2_ up to a lag of 21 days. We further conducted multi-pollutant models and stratified our analysis by age group, sex, and season. Results: The overall relative risk (95% confidence interval (CI)) for PM_10_, NO_2_, and SO_2_ at lag 0–1 for hospital admissions due to respiratory disease (RD) were 1.9% (0.5–3.2%), 2.3% (0.6–4%), and 1.1% (−0.2–2.4%), respectively. For cardiovascular disease (CVD), these values were 2.1% (0.6–3.5%), 1% (−0.8–2.8%), and −0.3% (−1.6–1.1%), respectively, per inter-quartile range increase of 12 µg/m^3^ for PM_10_, 7.3 µg/m^3^ for NO_2_, and 3.6 µg/m^3^ for SO_2_. The overall cumulative risks for RD per IQR increase in PM_10_ and NO_2_ for children were 2% (0.2–3.9%) and 3.1% (0.7–5.6%), respectively. Conclusion: We found robust associations of daily respiratory disease hospital admissions with daily PM_10_ and NO_2_ concentrations. Associations were strongest among children and warm season for RD.

## 1. Introduction

Over four million deaths are attributed to outdoor air pollution yearly, as reported by the World Health Organization (WHO); the majority of these deaths are largely cardiovascular and respiratory diseases [1]. Air pollution is the biggest threat to public health, of which the highest exposure is in low- and middle-income countries [2]. In these countries, the air quality levels are not compliant with the new WHO guideline values nor the previous ones [1,3].

Air pollution is composed of a complex mixture of gases and particulate matter. The most common pollutants are nitrogen dioxide (NO_2_), sulfur dioxide (SO_2_), ground-level ozone (O_3_), carbon monoxide (CO), and particulate matter (PM). PM are subdivided on the basis of their sizes: particles with 2.5 microns in diameter or less are PM_2.5_, while particles up to 10 microns in diameter are PM_10_. The mixtures and concentrations of these pollutants differ by countries. 

Several studies have demonstrated the short-term associations of ambient air pollutants with cardio-respiratory diseases and death. An Italian study reported a 10 µg/m^3^ increase in ambient NO_2_ was associated with 1.19% (0.23–2.15%) and 1.20% (0.17–2.23%) increases in respiratory disease (RD) and in chronic obstructive pulmonary disease (COPD) hospital admissions, respectively. In addition, a 10 µg/m^3^ increase in the daily mean of PM_10_ was associated with a 0.59% (0.10–1.08%) and 0.67% (−0.02–1.35%) rise in RD and COPD hospital admissions, respectively [4]. A Belgian study found an association between CVD hospital admissions and NO_2_ with an overall risk of 3.5% (2.4–4.7%) per 10 µg/m^3^ [5]. In addition, Liu et.al., conducted a study which involved 652 cities and found an 0.4% increased risk in all-cause mortality in South Africa per 10 µg/m^3^ of PM_10_ [6].

There is an insufficient number of studies on ambient air pollution (AAP) from sub-Sahara Africa (SSA), and the majority of air pollution studies are on indoor air pollution due to the burning of fuel for household use. In 2018, a systematic review reported only 12 studies from SSA that derived concentration-response functions (CRF) for any health outcome using ambient air pollution (AAP) measurements [7].

Therefore, the wealth of knowledge on the subject and the current evidence stems from Europe and North America. It is rather uncertain whether these results can be extrapolated to sub-Saharan African (SSA) countries. Air pollution levels are not only higher in the latter but also from different sources. In addition, effects of air pollution could be more strongly influenced by population-level vulnerability and individual-level susceptibility in SSA than in high income countries (HICs). These vulnerabilities include but are not limited to differences in socioeconomic risk factors, environmental conditions, or the co-occurrence of the double burden of chronic and infectious diseases such as tuberculosis (TB) and HIV/AIDS [7].

For instance, South Africa has a large double burden of diseases from both communicable diseases—HIV/AIDS and TB—and non-communicable diseases (NCDs) such as cardiovascular and chronic lung diseases (including asthma) [8]. The high infection rate could make people more vulnerable to inflammatory effects of ambient air pollution that sustain NCDs. Furthermore, in addition to these diseases being a risk for the development or worsening of cardiorespiratory diseases, South Africans are exposed to levels of air pollution that exceed the 2021 WHO guideline values and concentrations observed in Europe or North American countries [9].

In 2012, Wichmann et al. reported a 3% increase in daily CVD deaths per 12 µg/m^3^ and 8 µg/m^3^ contrasts in NO_2_ and SO_2_ daily means, respectively. These estimates are substantially larger than what was published in systematic reviews from other countries. A review from China found an increase of 0.75% and 1.12% for a corresponding rise of 10 µg/m^3^ in NO_2_ and SO_2_ for CVD deaths, respectively [10].

There have been a few studies to date in SA and SSA at large that have explored the short-term association between monitored air pollutants and any health outcomes [6,11,12,13], but none has quantified the joint health effects of multiple pollutants on hospital admissions for any health outcome.

Therefore, the objective of this study was to address this gap in knowledge by investigating the short-term associations between daily averages of three routinely monitored air pollutants (PM_10_, NO_2_, SO_2_) and daily hospital admissions due to cardiorespiratory disease in the City of Cape Town using a time-series analysis. In addition, we evaluated the difference in associations by sex, age, season, cumulative lag effects, and multi-pollutant models. This study is in line with the sustainable development goals (SDGs), particularly goals 3 and 13; furthermore, findings from this study can contribute to policymaking and the re-evaluation of South Africa’s air quality guidelines.

## 2. Methods

### 2.1. Study Area

Data from Cape Town (CT), South Africa, was used to conduct this study. CT is the second most populous city in SA, with an estimated population of 3.7 million residents, 69.6% being in the working age group (15–64 years). The population density is 1530 people per km^2^ in a total area of 2461 km^2^ with over 1 million households. It has a subtropical Mediterranean climate, where the winter (May–August) cold front comes from the Atlantic Ocean with heavy precipitation and strong northwesterly winds. The summer months are warm (September–April) and dry, with frequent strong winds from the southeast (Indian Ocean) and the north (semi-arid Karoo interior).

### 2.2. Hospital Admission Data

Daily counts of respiratory and cardiovascular diseases hospital admissions were obtained by age and sex from seven private hospitals from 1 January 2011 to 30 October 2016. The health outcomes were coded using the International Classification of Disease, 10th version (ICD−10) (J00-J99) and (I00-I99). Private hospital data were used at the time of the study because the City of Cape Town public hospital data were not electronically available.

### 2.3. Air Pollution Data

Daily hourly averages of PM_10_, NO_2_, and SO_2_ were obtained from the City of Cape Town air quality monitoring stations for the same study period. Ambient air quality has been monitored in the City of Cape Town since the 1960s with 14 stations in their network. PM_2.5_ was not included in the analyses as the South African standard only came into effect in 2012; thus, time series are shorter and less complete [14]. We used PM_10_, NO_2_, and SO_2_ data from January 2011 to October 2016; during this period, PM_2.5_ data were unavailable for some years and missing for others. These pollutants were measured, but not at all the stations; out of the 14 monitoring stations, 8 measured PM_10_ and 7 measured NO_2_ and SO_2_. A map with the locations of the monitoring stations and a description of annual measurements are provided in the Appendix A.

South Africa requires continuous monitoring of criteria air pollutants, as stated in the National Environmental Management Act: Air Quality Act in 2004 [15]. For each station, the daily means were derived if a minimum of 18 h (75%) of hourly data were available. To obtain a city-level daily mean, the daily average across all stations for days with measurements were calculated. We then took a step further by imputing measurements for stations with missing values to be additionally used in calculating the city-level daily average. This required the presence of at least one measurement at any of the remaining stations on the respective day. The respective algorithm is explained in detail in the online Appendix A. A different approach was applied in the prediction of missing PM_10_ data in a different study using monitoring stations from Cape Town. It was reported that models for each site performed better in capturing the variability of PM_10_ concentration [16].

### 2.4. Meteorological Data

Daily meteorological variables, which included temperature and relative humidity, were obtained from the South Africa Weather Service (SAWS), while wind direction, wind speed, and solar radiation were obtained from the European Centre for Medium-Range Weather Forecasts (ECMWF) re-analysis dataset [17]. 

### 2.5. Statistical Analysis

Descriptive daily statistics of hospital admissions, air pollutants, and meteorological variables were calculated for the entire study period, with data presented for the total population and after stratification by sex (male and female), age (years 0–14, 15–64, and ≥65), and seasons—warm (January–April and September–December) and cold (May–August). The temporal correlation between the air pollutants and meteorological variables was assessed using Spearman’s rank correlation coefficient.

The associations between daily mean concentrations of the air pollutants and daily counts of respiratory and cardiovascular hospital admissions were assessed in separate analyses using generalized additive quasi-Poisson models. This approach has been used in several time-series studies [18]; it uses several parameters to explain the contributions of different time lags of pollutant exposures to the total effect of the respective pollutant from within the respective time window on the daily number of events. Therefore, it provides a comprehensive picture of the time-dependency of the exposure–response relationship over different specified lags [19]. The core model included natural cubic spline functions of calendar time with a fixed number of yearly knots, 12 for respiratory diseases (RD) and 4 for cardiovascular diseases (CVD), to control for time trends and seasonal patterns in hospital admissions. In addition, indicator variables for public holidays as defined by the government and for the different days of the week were added. Furthermore, we created variables cost and sin t with periods of one year and added them to the models along with interaction terms with the day of the week indicator variables, in order to capture potential seasonal variations in the day-of-the week effects. These variables were defined as shown below:(1)$$cos\;t=cos\;\left(\frac{time\times 2\times \pi }{365.25}\right)$$
(2)$$sin\;t=sin\left(\frac{time\times 2\times \pi }{365.25}\right)$$

We controlled for effects of temperature and relative humidity with natural spline functions of their averages over lags 0 to 3. The primary pollutant variables were the two-day means over lags 0 and 1. RD and CVD had to be modelled slightly differently, as RD required more degrees of freedom per year than CVD. Autoregressive terms were added if we were unable to adequately remove partial autocorrelation at short lags.
(3)$$\begin{array}{ll} \log\left(E\left(CVD_{total}\right)\right) & =gam(CVD_{total}\;\sim \;pollutant_{lag_{0-1}}+ns(time,df\;\;\;\\ & =4\times 6)+s\left(\;temp{\;}_{lag_{0-3}}\right)+\;s\left(rh{\;}_{lag_{0-3}}\right)\;\;\\ & +\;as.factor\left(dow\right)\times cost+\;as.factor\left(dow\right)\times sint\;\;\\ & +as.factor\left(pubday\right),\;method=“REML”,family\\ & =“quasipoisson”,\;data=data,na.action=na.exclude) \end{array}$$
(4)$$\begin{array}{ll} \log\left(E\left(RD_{total}\right)\right)= & gam(\;RD_{total}\;\sim \;\;pollutant_{lag_{0-1}}\;+\;s(time,df\\ & =12\times 6)+s\left(\;temp{\;}_{lag_{0-3}}\right)+\;s\left(rh{\;}_{lag_{0-3}}\right)\;+\;as.factor\left(dow\right)\times cost\\ & +\;as.factor\left(dow\right)\times sint\;+\;as.factor\left(pubday\right),\;method\\ & =“REML”,family=“quasipoisson”,\;data=data,na.action\\ & =na.exclude) \end{array}$$

Residuals of lag 1 were added after the core model was derived, in order to remove existing lag1-autocorrelation of residuals. Single pollutant models were derived for PM_10_, NO_2_, and SO_2_ separately, followed by two- and three-pollutant models. To facilitate comparison of associations across pollutants, we presented the results as relative risks (RR) and 95% confidence intervals (CI) for an interquartile range increase in the respective pollutant variable. Results for a 10 µg/m^3^ increment are tabulated in the Appendix A.

In addition to the overall analysis, the models were stratified by sex (male vs. female), age groups (<15, 15–64, and >65), and seasons (warm vs. cold months). Statistical significance was defined as two-tailed *p*-value < 0.05. Chi^2^ tests were used to compare effect estimates across different subgroups. 

Finally, within a distributed lag non-linear model (DLNM) framework, we used a cross basis function for the lag model of each pollutant variable, which included a linear exposure–response function and a natural cubic spline for the lag weights. A total of 21 lags were considered and knots were placed at lags 2, 5, and 9. For temperature and relative humidity, we considered lags 0 to 3 and used argvar and arglag with natural spline and 5 degrees of freedom for the exposure–response relation. The respective equations in R are given below. The statistical analysis was performed using R software, version 4.0.3 (R Foundation for Statistical Computing), using the mgcv and dlnm packages for fitting the models.
(5)$$\begin{array}{c} poll_{cs}=crossbasis\;(poll_{conc},\;lag=21,\;argvar=list\left(fun=“lin”\right),\;arglag\;\\ =list(fun=ns,\;knots=c\left(2,5,9\right) \end{array}$$
(6)$$\begin{array}{c} \;\;\;met_{cs}=crossbasis(\;met,\;lag=3,\;argvar=list\;\left(fun=“ns”,df=5\right),\;\;\\ arglag=list\left(fun=“ns”\right)\;) \end{array}$$
where *poll_cons_* = daily pollutant concentration, *met* = daily average of meteorological variable, *lin* = linear, *ns* = natural cubic spline, *met* = daily temperature and relative humidity.

## 3. Results

During the study period 1st Jan 2011–31st Oct 2016, 54,818 cardiovascular (25.7 cases per day) and 58,317 (27.4 cases per day) respiratory disease hospital admissions were recorded, as described in Table 1.

The daily average of CVD admissions was similar in both warm and cold seasons; however, there were more admissions in the cold season than the warm season for RD, 33 compared to 24 per day. There were only 498 CVD admissions observed among ages 0–14 during the study period; this group was excluded from further analysis for CVD models.

The overall daily mean concentrations of PM_10_, NO_2_, and SO_2_ were 24.4 µg/m^3^, 15 µg/m^3^, and 9.4 µg/m^3^, respectively; average concentrations were similar for both warm and cold seasons, except for NO_2_. The highest PM_10_ level was 80.2 µg/m^3^ (Table 1), and PM_10_ exceeded the daily WHO 2021 guideline values of 45 µg/m^3^ on 123 days, NO_2_ exceeded the value of 25 µg/m^3^ on 237 days, and SO_2_ exceeded the guideline of 40 µg/m^3^ on 9 days. 

Daily temperature and relative humidity were 17.3 °C and 68.6%, respectively. Table 2 shows the Spearman correlation coefficients for daily air pollutants and meteorological variables. Correlations among pollutants were relatively low. For instance, PM_10_ and NO_2_ had a weak correlation of only 0.30. In the cold months, the latter correlation increased to 0.57, whereas in the warmer months, correlations between pollutants were lower, for instance, 0.19 for PM_10_ and NO_2_.

Table 3 shows the risk ratios and 95% confidence intervals of hospital admissions for respiratory and cardiovascular disease per interquartile range (IQR) of the 2 day moving average (lag 0–1) mean concentrations of the pollutants. PM_10_ showed a positive and statistically significant association with admissions for respiratory diseases, with an overall effect estimate of 1.9% (95%CI 0.5–3.2%) per IQR increase of 12 µg/m^3^. The strongest effect estimates were observed in age 0–14 years (2%, 95% CI 0.2–3.9%) and males (2%, 95% CI 0.2–3.7%) for the same unit. NO_2_ also showed a positive association, with an estimated increase by of 2.3% (0.6–4%) in RD admissions per IQR rise of 7.3 µg/m^3.^ The corresponding estimate was 3.1% (95% CI 0.7–5.6%) among children below the age of 15. The other groups also showed positive associations that did not reach statistical significance. However, we also observed a positive association with RD-admissions for SO_2_, with an estimate of 1.1% (−0.2–2.4%) per IQR increment of 3.6 µg/m^3^. The respective estimates were similar in all subgroups.

Cardiovascular disease (CVD) admissions increased with increasing PM_10_ and NO_2_ levels, but only PM_10_ risk estimates were statistically significant. In the unstratified analysis, an interquartile range increase in PM_10_ increased CVD hospitalizations significantly by 2.1% (0.6–3.5%). Statistically significant positive associations were observed in all groups. The overall percentage change in risk for CVD hospitalization associated with an interquartile increase of NO_2_ was 1% (−0.8–2.8%), while the respective estimate for SO_2_ was −0.3% (−1.6–1.1%). We observed stronger associations with RD than CVD hospital admissions for all three pollutants.

Results of single- and multi-pollutant models are presented in Figure 1. NO_2_-related effect estimates for respiratory diseases were largest (per IQR) across all models for this health outcome; the results were statistically significant for the single pollutant model and the two-pollutant model with SO_2_. In addition, a stronger association was observed among children (3.1%). In the multi-pollutant models, the overall percentage change for NO_2_ was 1.6% and 2.1% when adjusted for PM_10_ and SO_2_, respectively; results are found in the Appendix A. The estimate for NO_2_ decreased after the inclusion of PM_10_ and was lowest in the three-pollutant model with 1.5%. Similarly, the PM_10_ estimate was reduced from 1.9% to a non-significant value of 0.9% after adjusting for NO_2_, while SO_2_ effect estimates were statistically insignificant for all ages, groups, and models. In CVD, associations per IQR were strongest for PM_10_, and the respective RR estimates were not sensitive to adjustment with NO_2_ and SO_2_. The group-specific risk of PM_10_ on CVD hospitalization increased from 2.2% and 2.1% in a single pollutant model for age ≥ 65 years and males to 2.6% when adjusted for SO_2_, respectively; this is reported in the Appendix A. Associations for NO_2_ and SO_2_ with CVD hospitalizations were not statistically significant. 

Season-specific associations for respiratory diseases in single pollutant models of PM_10_, NO_2_, and SO_2_ are presented in Figure 2. PM_10_ and NO_2_ showed stronger associations in the warmer months (Sep–Apr) compared to the colder months (May–Aug) for CVD. However, the differences between seasonal estimates were not statistically significant. 

In the warmer months, respiratory disease (RD) admissions were positively associated with only PM_10_, with estimates of 2.2% (95% CI 0.5–3.9%) per IQR rise in lag0–1 of PM_10_. For an IQR rise in the two-day moving average of NO_2_, there was a risk of 2.3% (−0.4–4.9%), and a similar null finding in the two-day moving average of SO_2_, an IQR increase corresponding with a relative risk of 1.3% (−0.4–3.1%). 

Daily CVD hospital admissions were positively associated with PM_10_ and NO_2_ in the warmer months. An IQR increase in the two day moving average of PM_10_ corresponded with an increased risk of 3.4% (1.5–5.3%), which was the strongest association observed for all three pollutants. NO_2_ showed an increased risk of 2.7% (95% CI 0.2–5.4%) for an IQR increase in the two day moving average, while SO_2_ had a negative association with a risk ratio of −0.8% (95% CI −2.4–0.9%). In the colder months, there were no statistically significant associations for all the pollutants.

In the colder months, only NO_2_ showed any statistically significant association. The relative risk estimate for RD hospital admissions associated with an IQR increase in the two day moving average of NO_2_ was 2.3% (95% CI 0.1–4.6%), while the corresponding estimates for PM_10_ and SO_2_ were 1.3% (95% CI −0.4–3.3%) and 0.8% (95% CI −1.2–2.9%), respectively.

### Lag Models

Figure 3 presents the estimated lag structure of the effects of a 10 µg/m^3^ increase in PM_10_, NO_2_, and SO_2_ concentration on respiratory disease admissions for the whole group. Age-specific lag structures for PM_10_ are illustrated for ages 0–14 and ages ≥ 15 years; we noted from Table 3 that ages 0−14 appeared to be driving the estimates for PM_10_, and this was confirmed by the graphs. This age group showed both acute (lag 0–1) and delayed (lag 6–7) associations in comparison to the age group 15 years and older. Conversely, for NO_2_, we observed no acute effect at lag 0–1 for the whole group, even though the two day moving average was significant, whereas SO_2_ showed significant negative effect from lags 8 to 18. We tested for differences in estimates between age groups and sex using chi-squared tests but found no statistically significant differences.

When we analyzed the effect of a 10 µg/m^3^ increase in PM_10_ on cardiovascular disease admissions, as presented in Figure 4, we saw a similar pattern to RD; however, in this case, the effect was observed in all groups, as indicated in Table 3. There were acute effects from lags 0–1 and delayed effects from lags 5–9. For NO_2_, lag 0 was the only statistically significant contributor to CVD hospitalizations as the confidence interval for lag1 reached below 1, while we did not see any significant contributions from SO_2_ across all lags. In addition, when the data were stratified by age and sex, we observed no association for each respective pollutant on CVD admissions. 

## 4. Discussion

This is the first study in sub-Sahara Africa to investigate the association between multiple air pollutants and hospital admission due to cardiovascular (CVD) and respiratory diseases (RD). Associations for RD hospital admissions were consistently observed with PM_10_ and NO_2_ in both single- and SO_2_-adjusted models; however, for CVD, only PM_10_ showed positive associations. In contrast to various other studies, the temporal correlations between the three pollutants were rather low; thus, our study had the opportunity to investigate the independent associations of each pollutant. We did not find statistically significant associations between SO_2_ and hospital admissions of RD and CVD. 

Our findings for PM_10_ are in line with the current knowledge about the ability of fine particles to penetrate into the respiratory tract and the lungs and affect the heart. Numerous studies have demonstrated that exposure to air pollution is followed by an oxidative stress reaction initiated by inflammatory response to PM entering the lung [20,21]. The oxidative reaction from the lung is further amplified through a different enzymatic pathway finally leading to a systemic vascular oxidative stress reaction [22]. Systemic inflammatory responses have been demonstrated in both animal and controlled human studies [23]. The particle dose and composition determines the extent of the pulmonary inflammation; controlled human exposure studies have demonstrated increased markers for pulmonary inflammation for exposure to a variety of particle types [24].

However, in comparison to other studies, for a 10 µg/m^3^ increase in PM_10_, our estimates of 1.8% for RD and 1.9% for CVD were higher than the pooled estimates reported globally and from low- and middle-income countries (LMIC) for cardiorespiratory morbidities. Newell et al. conducted a meta-analysis and systematic review of association between PM_10_ per 10 µg/m^3^ increase across a 0–1 day lag and cardiorespiratory outcomes in 39 studies and reported a RR(CI) of 0.39% (−0.04–0.8%) in daily RD hospitalization [25]. In Switzerland, a 10 µg/m^3^ increment in PM_10_ was associated with increased risks for RD and CVD daily admissions by 0.22% (95% CI: −0.43 to 0.87%) and by 0.43% (95% CI: 0.12–0.73%), respectively [26]. However, similar findings with stronger associations were observed for RD and CVD mortality in a study based on Cape Town data [11].

The larger association with PM_10_ observed in our study might be explained by a number of reasons. The maximum concentrations of 80.2 µg/m^3^, 42 µg/m^3^, and 23.2 µg/m^3^ for PM_10_, NO_2_, and SO_2_, respectively, were quite comparable to what has been reported in other European studies [27], given the advantageous coastal location of Cape Town. 

Firstly, the difference in sources, composition, and toxicity of the pollutant with respect to the study area might play a role. In SA, residents are exposed to air emissions from landfill sites, tires and open refuse burning, airports, agricultural activities, windblown dust, and transboundary air pollution [28]. Furthermore, the City of Cape Town is prone to structural and vegetation fires. Yearly, the city deals with an average of 8600 vegetation fires and more than 100 fires from informal settlements as a result of accidents with paraffin stoves [29]. Consequently, these sources of exposure will have an influence on the chemical composition of PM_10_ with possibly different constituents including more crustal materials. Research has shown that during high pollution days in Cape Town, most of the air parcels have travelled from major dust source regions such as the Kalahari and Namib deserts before arriving over the city. In addition to favorable atmospheric conditions for the dispersion of air pollution, peaks in PM_10_ concentrations are associated with transport of PM_10_ plume driven by northerly flow induced by coastal and continental high pressure systems [30]. Another study from Cape Town investigated the sources and chemical composition of PM_2.5_ and soot levels, finding significant correlations with PM_10_ levels measured 3 km from one of the monitoring stations used in this study. Cl^−^, NO_3_^−^, SO_4_^2−^, Al, Ca, Fe, Mg, Na, and Zn were detected in the samples collected for 121 days, of which the largest fraction of the PM_2.5_ samples were due to anionic and metallic species. If those levels are assumed to be representative for the city, one may observe higher risk estimates [31]. Secondly, these stronger effects may also represent difference in vulnerability and high prevalence of pre-existing diseases such as hypertension and atherosclerosis, which increases their susceptibility to stronger acute effects [32].

### 4.1. Overall Association of Hospital Admissions and NO_2_

We found statistically significant associations between NO_2_ and RD hospitalizations but not with CVD. For both health outcomes, the observed point estimates were higher in our study than reported by others. For instance, a systematic review from 2015 using estimates from 18 WHO regions reported 0.57% (95% CI 0.33–0.82%) and 0.66% (95% CI 0.32–1.01%) increases in RD and CVD hospital admissions, respectively [33], while our respective estimates were 3.4% and 1.7% (per 10 µg/m^3^ ).

A meta-analytic study investigating the association between NO_2_ and hospital admissions from Italy’s most polluted region showed risks of 1.20% (90% CI 0.53–1.81%) and 1.14% (90% CI 0.51–1.83%) for RD and CVD admissions, respectively [34]. Sunyer et al. reported a 0.7% (95% CI: 0.1–1.3%) increase in CVD daily admissions per 10 µg/m^3^ of NO_2_ from seven European areas. However, another systematic review [25] from 22 studies reported increased risks ranging from 1.08% to 1.94% and 1.04% to 1.17% for RD and CVD hospitalization, respectively [35]. 

The major source of NO_2_ is vehicle emissions, especially in urban areas where they may be responsible for 60–70% of nitrogen oxides in the atmosphere [28]. This is the case for areas with high traffic density such as central business districts. If the mix of traffic-related pollutants in South Africa is different from those observed in Europe, this might, at least in part, explain the higher effect estimates for NO_2_ observed in our study. The current emission legislation in SA is equivalent to Euro II (Euro 2) only. Although the importation of used vehicles is banned, the overall car fleet is still not at the level found in Europe. Moreover, the fuel quality is much worse than in Europe, which may substantially affect the quality and toxicity of the emissions beyond NO_2_. In 2018, the maximum sulfur limits in gasoline were 501–3500 parts per million (ppm), while road diesel had limits of 351–500 ppm, as compared to the much lower European limits of 5–10 ppm [36]. Indeed, as shown in Figure 2, adjustment for ambient PM_10_ substantially reduced the associations with NO_2_ to 1.6% for RD. This may also point in the direction of NO_2_ being a marker for rather complex combustion-related constituents, which in turn are partly captured by particles.

### 4.2. Overall Association of Hospital Admissions and SO_2_

Our study showed that daily concentrations of SO_2_ were not significantly associated with RD nor CVD hospitalizations; these results are similar to findings from previous studies. A study conducted in England and Wales reported no association with CVD admissions with a risk of −0.3% (95% CI −1.0–0.3%) per 10th−90th centile increment in SO_2_ (10.4 µg/m^3^) from lag 0–4 days [37]. The median concentration 3.1 µg/m^3^ (2–6) in this study was smaller in comparison to our study (see Table 1).

In addition, a systematic review of low- and middle-income countries (LMIC) found an association between SO_2_ exposure and cardiorespiratory morbidity. At lag 0–1 with 10 µg/m^3^ rise in SO_2_, the risk for RD was increased by 0.40% (95% CI 0.19–0.61%), while CVD, as in our data, was not associated with 10 µg/m^3^ increase in SO_2_ (0.07% 95% CI −0.40, 0.55%) [38]. In our multipollutant models, SO_2_ showed consistent positive but non-significant associations with RD hospitalizations and no associations with CVD hospitalizations, whereas effect estimates for NO_2_ and PM_10_ on RD and CVD hospitalizations slightly increased when they were adjusted for SO_2_. 

SO_2_ comes from burning of coal and oil, and SA generates more than 90% of its energy from coal burning. As a gaseous compound, the emission of SO_2_ can lead to the formation of secondary particles while also acting as surrogate for other substances. According to the United States Environmental Protection Agency (US EPA), more than 12% of SO_2_ emitted in urban areas are converted in the atmosphere to sulphate particulate matter [39]. Furthermore, there is the issue of transboundary pollution from other parts of the country and neighboring countries. Concurrently, there are other sources of pollution such as open waste burning and emission from traffic that is worsened by the high sulfur content in fuel for on-road vehicles and industry. This results in a complex mixture of pollutants, making it difficult to disentangle their associations with health outcomes.

### 4.3. Effect Modification by Age Group, Sex, and Season

In our study, we found the highest estimates for PM_10_ and NO_2_ on RD hospitalizations in children. This finding is in line with other studies on cardiorespiratory diseases showing greater sensitivity of the youngest to air pollution [40,41,42,43,44,45]. However, the elderly also showed slightly higher estimates.

Individual response to air pollution exposure can be determined by several factors, which include pre-existing diseases, socio-economic status, age, and lifestyle. However, children are particularly vulnerable to the adverse effects of air pollution, and as reported by the WHO [44], more than 90% of children are exposed to toxic air, particularly in low- and middle-income countries. Air pollution can also alter the development and function of a child’s lungs as they are still growing [46]. In addition, children spend more time outdoors and engage in physical activity, which increases their breath rate and allows for environmental pollutants to be deposited in their respiratory tract in larger amounts [47]. Furthermore, children are predominantly oral breathers, which means more polluted particles enter their lower airways as a result of their nasal filter being by-passed [48]. 

One possible cause of increased sensitivity to air pollution of the elderly is the higher prevalence of co-morbidities that reduce biological functions or resistance to infection and inflammatory ailments [49]. The human body experiences physiological degeneration with increased age. The normal function of the body organs is affected by aging, which results in cardiovascular, urinary, and respiratory health conditions [50]. Consequently, the ability of older people to adapt to increased concentrations of air pollution and changing weather conditions is reduced [51]. Moreover, in comparison to young people, older people have lower immunity and antioxidant defense and more progressed atherosclerosis, which puts them at a higher risk for acute effects of air pollution [52].

Gender-specific estimates did not differ significantly; therefore, this may just reflect random variation. This is in line with a meta-analysis where 13 studies out of 14 on associations between short-term exposure to air pollution and cardiorespiratory disease hospital admission did not find statistically significant effect modification by sex [53]. 

We found stronger associations of air pollution with cardiovascular hospitalizations in warmer months than in colder months. In RD and CVD, the overall effect estimates for a 10 µg/m^3^ increase in PM_10_, NO_2_, and SO_2_ were considerably larger in the warmer than in the colder months. Similar findings were observed in an earlier study that investigated temperature as a modifier of the effects of air pollution on CVD hospitalizations in Cape Town. The study reported stronger associations in warmer months compared to colder months [12]. The higher and significant effect estimates observed in warmer months might be explained by the time spent outdoors for leisure and physical activities. Increased duration of exposure and increased respiratory rate due to outdoor activities both contribute to higher doses of inhaled air pollution. In addition, wildfires are more frequent in summer than winter, particularly from burning of waste at landfill sites. This will have an influence on the characteristics of the pollutants and possibly the level of toxicity of the mixture of pollutants. During warmer months, oxidant pollutants such as ozone and other secondary pollutants are also higher, which may amplify the effects of pollution. Furthermore, temperature and humidity drive the dispersion of air pollution differently in both seasons. For instance, humidity is higher in winter than in summer, which reduces the distribution of air pollution in winter.

### 4.4. Strength and Limitations

This study has some limitations; firstly, misclassification of air pollution exposure could be a potential source of bias in our study as it is likely that, in an urban city such as Cape Town, there will be substantial spatial heterogeneity of these concentrations. Although all monitoring stations in the city were used to calculate daily average ambient pollutant concentrations, these will differ to some extent from the population-averaged daily exposure levels, which would ideally be used in time series analyses of hospital admission. However, population-averaged exposure estimates would have required models of daily pollutant levels of high spatial resolution, which were not available. On the other hand, it has been reported that the temporal variation at fixed monitoring sites is well correlated with the temporal variation of personal exposure to particulate matter, which should reduce random misclassification [54,55]. However, high correlation between individual exposure and fixed site levels does not imply that there will be little bias in the estimate.

Secondly, we did not include ozone in the analysis because there was insufficient data. Ozone may not only affect health but could potentially confound or modify the associations seen with other pollutants, especially during the warmer months when ozone levels are high. 

Thirdly, NO_2_ and SO_2_ had a considerable amount of data missing due to issues such as power failure, faulty equipment, and relocation of air quality monitors. However, we do not expect the pattern of missingness to be systematically associated with important factors such as toxicity and source of pollution. Therefore, this is unlikely to be a major source of bias for our estimates.

Lastly, the health outcome data were obtained from private hospitals, being part of the private health sector, which covers only 16% of the population [56]. People who can afford to pay out of pocket and those with health insurance have access to these hospitals, which are most likely people with middle and higher socio-economic status (SES). Therefore, generalizing of our effect estimates to the entire population of Cape Town and South Africa will need some caution. In addition, at the time of the study, public hospital data were not readily available electronically; however, the South African government has developed strategies to establish data electronic data sharing agreement between government entities and third-party users [57].

We would expect SES to modify the effect estimates of air pollution on cardiorespiratory health in Cape Town. People with lower SES are more likely to live in poorer and disadvantaged neighborhoods with less delivery of public services such as housing, electricity, and education, as well as less access to healthcare, healthy diet, water, and good roads. This could lead to increased susceptibility to the effects of air pollution combined with higher levels of exposure as there is a higher likelihood of them living closer to the roadways with increased traffic density and to more polluted areas. People with disadvantaged SES are also likely to live in houses with poor ventilation or increased indoor air pollution due to burning of wood and coal to generate heat. In addition, in the past, South Africa had a poor land use planning that resulted in heavy industrial developments in proximity to highly populated residential areas; residents of these areas are mostly people with lower and middle SES, and they may be exposed to higher levels of air pollution than those at monitoring stations. Furthermore, people with higher SES may live in areas with lower air pollution exposure and more green space. Moreover, they may have reduced susceptibility to air pollution effects in comparison with people with lower SES. In addition, the duration, frequency, and intensity of these exposures can contribute to variations in the magnitude of associations between different socio-economic groups.

A systematic review on stroke found that there was a stronger association with pooled estimates from low- and middle-income countries (LMIC) for NO_2_ and PM_10_ in comparison with high-income countries (HIC). They found higher median pollutant concentrations for both PM_10_ and NO_2_ in LMICs compared to HICs [58]. An Italian study reported a linear correlation between high levels of NO_2_ and PM_10_ concentrations and the lack of possession of a home, low level of education, and population density [59]. Numerous studies have investigated the influence of SES on adverse health effects of air pollution, and there have been conflicting reports of effect modification [60,61]. This makes it difficult to draw a definitive conclusion on whether the short-term effects of air pollution on health are modified by SES. Therefore, it is important that future studies use public hospital data to understand the influence of exposure to ambient air pollution on cardiorespiratory health for different sub-groups in a South African context.

## 5. Conclusions

In conclusion, PM_10_ and NO_2_ were found to be associated with RD, particularly in children, and with CVD. Associations were stronger in the warmer months. Our estimates are higher than what has been reported in North American and European studies, despite similar concentration levels. This may be due to the different sources and characteristics of the pollutants in SA. Given that we found stronger associations at relatively low levels and in light of the updated WHO air quality guideline values, it is crucial that interventions to reduce air pollution are implemented in South Africa, as this could potentially reduce the risk of morbidity, particularly for children and the elderly. Therefore, further investigation is needed, whilst health-based air quality standards should be adopted and enforced to protect public health.

## Figures and Tables

**Figure 1 ijerph-19-00495-f001:**
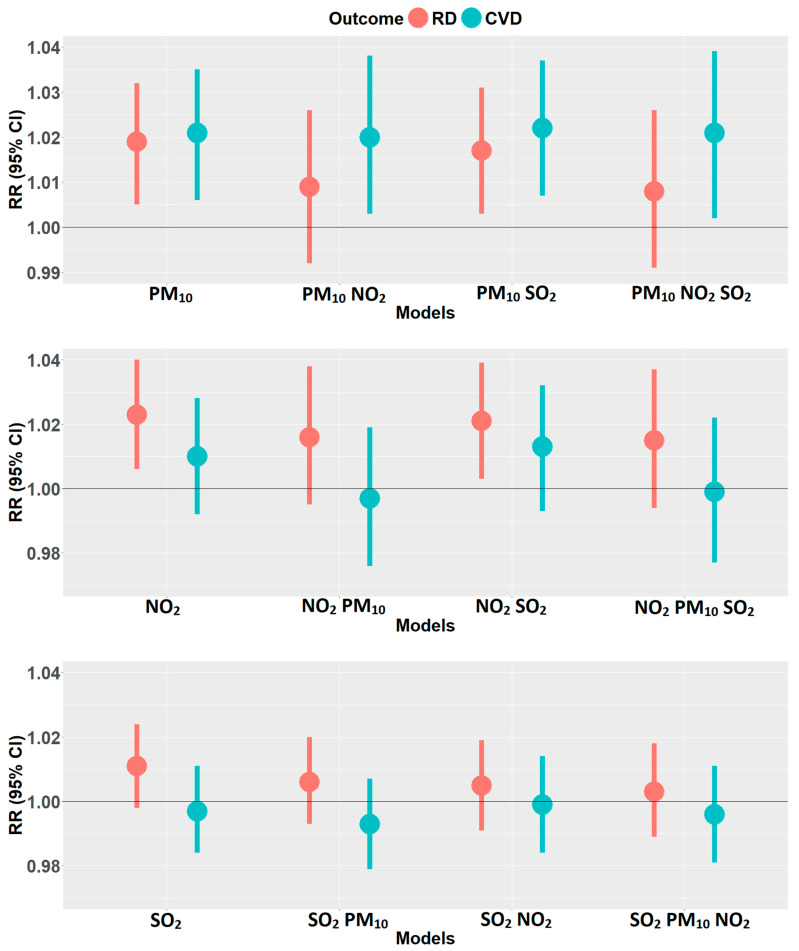
Estimated overall (lag 0–1) relative risks (with 95% confidence intervals) for respiratory (RD) and cardiovascular disease (CVD) admissions, per one interquartile range increase in the two day moving averages of PM_10_ (**top panel**), NO_2_ (**middle panel**), and SO_2_ (**bottom panel**) for single and multipollutant models. Sex- and age-specific estimates are reported in the Appendix A.

**Figure 2 ijerph-19-00495-f002:**
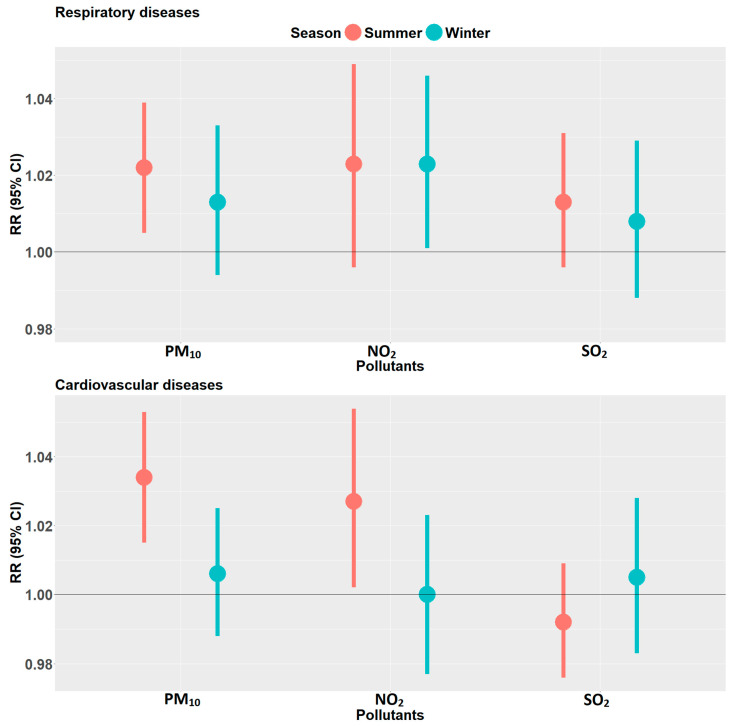
Season-specific relative risks of RD and CVD hospitalizations per interquartile range increase in the two day moving average (lag 0–1) of the respective pollutant. Summer season (January–April and September–December) and winter season (May–August), for respiratory (**upper panel**) and cardiovascular (**lower panel**) diseases, with bars representing 95% confidence intervals.

**Figure 3 ijerph-19-00495-f003:**
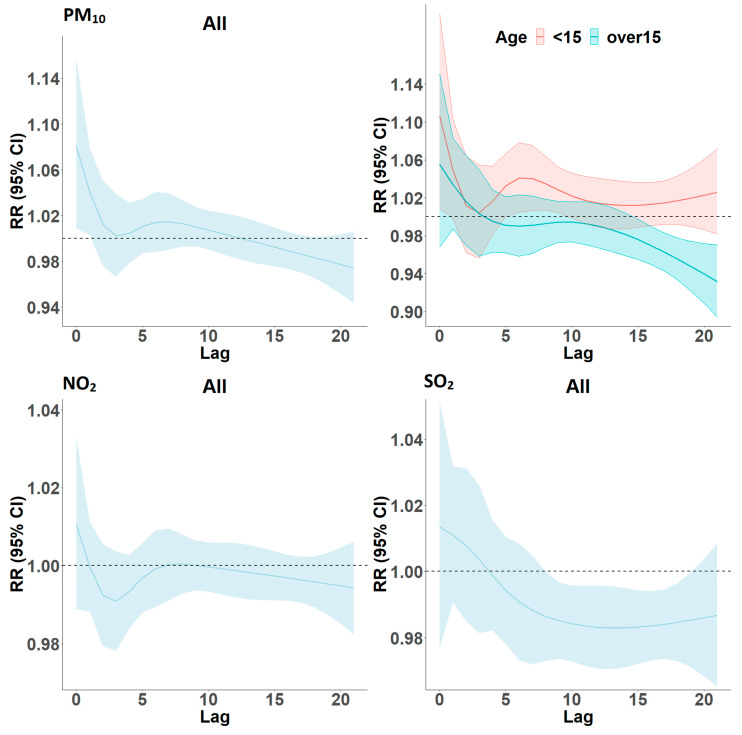
Estimated lagged effects (over 21 days) of PM_10_, NO_2_, and SO_2_ on respiratory disease hospital admissions, overall, and by age group for PM_10_ in Cape Town, South Africa, 2011–2016. Effect estimates are per 10 µg/m^3^ increment of the respective pollutant.

**Figure 4 ijerph-19-00495-f004:**
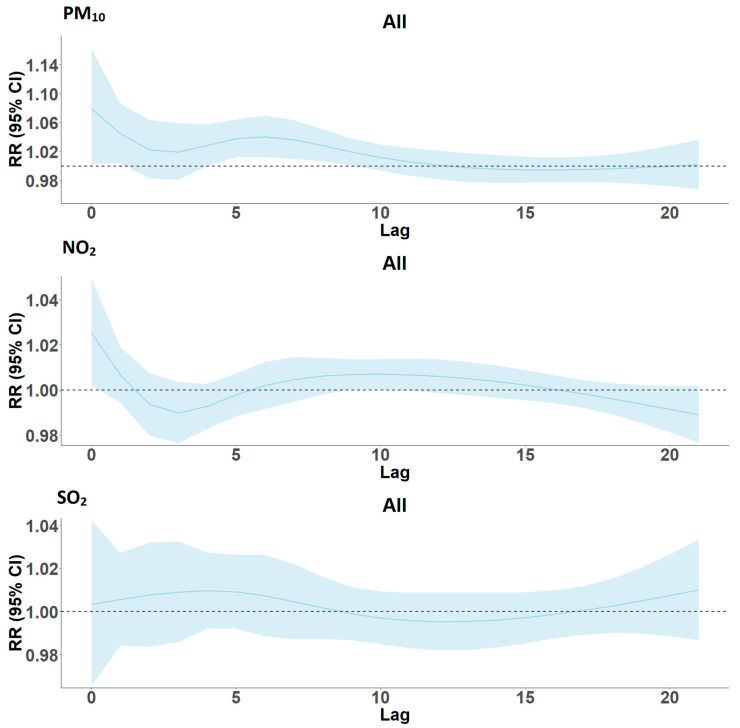
Lag structure (0–21) of the effects of a 10 µg/m^3^ increase in PM_10_, NO_2_, and SO_2_ concentrations on cardiovascular disease hospital admissions in Cape Town, South Africa, 2011–2016.

**Table 1 ijerph-19-00495-t001:** Summary statistics of daily number of cardiovascular and respiratory disease hospital admissions, daily mean concentrations of ambient air pollutants, and meteorological conditions in Cape Town, South Africa, from 1 January 2011 to 31 October 2016.

						Percentiles			By Season–Mean (SD)	
Variable	Mean	SD	Min	Max	IQR	25th	50th	75th	Warm	Cold
Cardiovascular disease									n = 35,487	n = 19,331
All ages and sex n = 54,818	25.7	13	2	66	23	13	27	36	25.5 (13.2)	26.2 (12.6)
0–14 years n = 498	0.2	0.5	0	3	0	0	0	0	0.2 (0.5)	0.2 (0.5)
15–64 years n = 27,225	12.8	7	0	33	12	6	13	18	12.8 (7.1)	12.7 (6.9)
>65 years n = 27,095	12.7	6.9	0	43	11	7	12	18	12.4 (7)	13.3 (6.8)
Female n = 22,914	10.8	5.9	0	35	10	5	11	15	10.6 (6)	11.1 (5.9)
Male n = 31,904	15	8.1	1	40	13	8	15	21	14.9 (8.2)	15.1 (7.8)
Respiratory disease									n = 33,840	n = 24,477
All ages and sex n = 58,317	27.4	13.1	1	75	21	16	27	37	24.3 (12)	33.2 (13)
0–14 years n = 28,518	13.4	7.4	0	37	11	7	13	18	11.8 (7)	16.3 (7.3)
15–64 years n = 19,418	9.1	5.4	0	32	8	5	9	13	8.2 (5.1)	10.9 (5.5)
>65 years n = 10,381	4.9	2.9	0	17	4	3	4	7	4.3 (2.6)	6 (3.1)
Female n = 29,741	14	7.2	0	44	11	8	13	19	12.2 (6.6)	17.2 (7.3)
Male n = 28,576	13.4	6.9	0	40	10	8	13	18	12.1 (6.5)	16 (6.9)
Air pollutants										
PM_10_ (µg/m^3^)	24.4	9.5	6.9	80.2	12	17.3	22.7	29.3	24.1 (8.7)	25 (10.7)
NO_2_ (µg/m^3^)	15	5.5	3.9	42	7.3	10.9	14.1	18.2	13.5 (4.8)	17.8 (5.8)
SO_2_ (µg/m^3^)	9.4	2.8	2.6	23.2	3.6	7.3	9	10.9	9.3 (2.9)	9.6 (2.8)
Meteorological data										
Temperature (°C)	17.3	4.3	7.5	29.3	7	13.8	17	20.8	19.4 (3.6)	13.5 (2.3)
Relative humidity (%)	68.6	10.5	30.7	100	15	61	69	76	65.6 (9.4)	74.2 (10.1)

Abbreviations: SD—standard deviation; Min—minimum; Max—maximum; IQR—interquartile range. Warm period: January to April and September to December; cold period: May to August.

**Table 2 ijerph-19-00495-t002:** Spearman’s rank correlation between city-level daily mean PM10, NO_2_, SO_2_, and meteorological parameters during the period of 1st Jan 2011 to 31 Oct 2016 in Cape Town, South Africa.

	PM_10_	NO_2_	SO_2_	Temperature	Humidity
PM_10_	1				
NO_2_	0.30	1			
SO_2_	0.20	0.27	1		
Temperature	0.23	−0.38	0.01	1	
Humidity	−0.29	0.02	−0.11	−0.39	1

**Table 3 ijerph-19-00495-t003:** Overall relative risk estimated from quasi-Poisson regression models of respiratory and cardiovascular disease hospitalizations, adjusting for time trends and seasonal variation, day of the week, public holiday, and meteorological factors including temperature and relative humidity. Estimates are presented per interquartile range (shown in Table 1) increase in the 2 day moving average (lag 0–1) of PM_10_, NO_2_, and SO_2_ concentrations for respiratory and cardiovascular disease hospital admissions for different age groups and sexes.

**Respiratory Disease Hospitalization by Pollutants**									
	**Per 12 µg/m^3^ PM_10_**			**Per 7.3 µg/m^3^ NO_2_**			**Per 3.6 µg/m^3^ SO_2_**		
**Groups**	**RR**	**95% Confidence Interval**		**RR**	**95% Confidence Interval**		**RR**	**95% Confidence Interval**	
		**Lower**	**Upper**		**Lower**	**Upper**		**Lower**	**Upper**
All	1.019	1.005	1.032	1.023	1.006	1.04	1.011	0.998	1.024
Age 0–14	1.02	1.002	1.039	1.031	1.007	1.056	1.015	0.997	1.033
Age 15–64	1.009	0.988	1.03	1.003	0.976	1.03	1.011	0.99	1.032
Age ≥ 65	1.019	0.994	1.046	1.005	0.972	1.039	0.989	0.963	1.015
Female	1.014	0.997	1.032	1.013	0.991	1.036	1.006	0.989	1.023
Male	1.02	1.002	1.037	1.019	0.997	1.042	1.015	0.998	1.032
**Cardiovascular Disease Hospitalizations by Pollutants**									
	**Per 12 µg/m^3^ PM_10_**			**Per 7.3 µg/m^3^ NO_2_**			**Per 3.6 µg/m^3^ SO_2_**		
**Groups**	**RR**	**95% Confidence Interval**		**RR**	**95% Confidence Interval**		**RR**	**95% Confidence Interval**	
		**Lower**	**Upper**		**Lower**	**Upper**		**Lower**	**Upper**
All	1.021	1.006	1.035	1.01	0.992	1.028	0.997	0.984	1.011
Age 15–64	1.021	1.002	1.039	1.006	0.983	1.03	0.998	0.98	1.015
Age ≥ 65	1.022	1.004	1.041	1.017	0.994	1.041	0.995	0.978	1.012
Female	1.02	1.001	1.04	1.007	0.983	1.033	1.009	0.99	1.028
Male	1.021	1.003	1.038	1.015	0.993	1.037	0.987	0.971	1.003

## Data Availability

Exposure data are available for download on the South African Air Quality Information System (SAAQIS) https://saaqis.environment.gov.za/; (accessed on 22 April 2019) however, restrictions apply to the health outcome data.

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
