# Peer review of "Short-Term Joint Effects of PM10, NO2 and SO2 on Cardio-Respiratory Disease Hospital Admissions in Cape Town, South Africa"

_ijerph, 2022, doi:10.3390/ijerph19010495_

Round 1

Reviewer 1 Report

Please refer to the attachment file.

Author Response

Comments from Reviewer 1

This manuscript is interesting, and it can be seen that the authors have done a lot of work. However, before publication, there are still some issues that need further improvement. Here are some of my suggestions for authors and editors to refer to:

Comment 1: Regarding the structure of the article, I think this article will show your research results and analysis more intuitively and clearly if you write Results and Discussion together.

Response: The authors are confident that the current structure which is in line with the journal’s requirement intuitively and clearly presents our findings. From a scientific perspective we also strongly prefer the clear distinction between the results of our analyses and its interpretation (Discussion).

Comment 2: Regarding the content of the article, it is recommended to add the word South Africa in the title, It’s better to replace "multiple pollutants" with the names of the three specific pollutants mentioned in the paper

Response: Yes, we have modified the title accordingly to reflects this recommendation

Comment 3: As far as I know, pm2.5 has a high risk of disease and is not worth neglecting. So, why this research did not do PM2.5 research, please give an explanation.

Response: We agree with this and we have provided an explanation in the method section of the article in line 112 – 117. There are no sufficient data for PM2.5 yet available. Fortunately, temporal correlation between PM10 and PM2.5 is also high in this part of the word, thus, findings for PM2.5 are not expected to be much different.  

Comment 4: Please continue to conduct literature research, and introduce and summarize the research progress in more detail for the relevant research in the SSA area.

Response: Thank you for this suggestion, we conducted a new literature search on epidemiological findings of multiple air pollutants in SSA and there were no recent literature relevant to our study.

Comment 5: Please add an explanation of the significance of this research at the end of the Introduction section.

Response: Further explanation has been added to the end of the introduction section

Comment 6: Supplement the map of the study area and the geographic location of the air quality monitoring station in 2.1 and 2.3.

Response: We agree with this and have incorporated your suggestion in section 2.3 by referencing the map in the supplementary material.

Comment 7: Please further modify some formatting problems and grammatical errors to make the reader's reading experience better.

Response: We have tried to correct all spellings, formatting and grammatical errors 

Reviewer 2 Report

This article focuses on current and important issues highlighting the relationship between the presence of air pollution and the number of hospital admissions. Numerous studies have shown that air pollution causes a wide variety of health consequences. These can be more or less life-threatening. The effects range from respiratory irritation to systemic diseases such as hypertension and premature death due to stroke or sudden cardiac arrest.
I agree with the authors that one of the more troublesome human and environmental manifestations of ambient air pollution is the accumulation of pollutants in the ground layer. The article has an adequate theoretical basis, relevant information and analysis, good partial (in the article) and final (in the conclusion) conclusions. The article uses original research by the author and cited research by other researchers, which enriches its content. It is written in a good language and is based on the analysis of current and well-chosen literature, although it could be further enriched with other positions, e.g.

  1. Czechowski P. O., Dąbrowiecki P., Oniszczuk-Jastrząbek A., Bielawska M., Czermański E., Owczarek T., Rogula-Kopiec P., Badyda A., A preliminary attempt at the identification and financial estimation of the negative health effects of urban and industrial air pollution based on the agglomeration of Gdańsk, Sustainability, 12, 42-59, MDPI 2020.
  2. Badyda A., Grellier J., DÄ…browiecki P.: Ambient PM2.5 Exposure and Mortality Due to Lung Cancer and Cardiopulmonary Diseases in Polish Cities. Advances in Experimental Medicine and Biology, Volume 944, pp. 9-17, 2017.
  3. Badyda A., Dabrówcki P., Czechowski P., Majewski G.: Risk of bronchi obstruction among non-smokers-Review of environmental factors affecting bronchoconstriction, Respiratory physiology & neurobiology, 209, 39-46, 04.2015. 

The research models were applied correctly. Systematics of models is not a simple issue, as the diversity of model types is mainly due to their precisely defined purpose. The article should therefore be regarded as an interesting introduction to a very important issue and treated as a scientific article.

Author Response

Comments from reviewer 2

Comment 1: This article focuses on current and important issues highlighting the relationship between the presence of air pollution and the number of hospital admissions. Numerous studies have shown that air pollution causes a wide variety of health consequences. These can be more or less life-threatening. The effects range from respiratory irritation to systemic diseases such as hypertension and premature death due to stroke or sudden cardiac arrest.
I agree with the authors that one of the more troublesome human and environmental manifestations of ambient air pollution is the accumulation of pollutants in the ground layer. The article has an adequate theoretical basis, relevant information and analysis, good partial (in the article) and final (in the conclusion) conclusions. The article uses original research by the author and cited research by other researchers, which enriches its content. It is written in a good language and is based on the analysis of current and well-chosen literature, although it could be further enriched with other positions, e.g.

  1. Czechowski P. O., Dąbrowiecki P., Oniszczuk-Jastrząbek A., Bielawska M., Czermański E., Owczarek T., Rogula-Kopiec P., Badyda A., A preliminary attempt at the identification and financial estimation of the negative health effects of urban and industrial air pollution based on the agglomeration of Gdańsk, Sustainability, 12, 42-59, MDPI 2020.
  2. Badyda A., Grellier J., DÄ…browiecki P.: Ambient PM2.5 Exposure and Mortality Due to Lung Cancer and Cardiopulmonary Diseases in Polish Cities. Advances in Experimental Medicine and Biology, Volume 944, pp. 9-17, 2017.
  3. Badyda A., Dabrówcki P., Czechowski P., Majewski G.: Risk of bronchi obstruction among non-smokers-Review of environmental factors affecting bronchoconstriction, Respiratory physiology & neurobiology, 209, 39-46, 04.2015. 

The research models were applied correctly. Systematics of models is not a simple issue, as the diversity of model types is mainly due to their precisely defined purpose. The article should therefore be regarded as an interesting introduction to a very important issue and treated as a scientific article.

Response: We thank the reviewer for the suggestions and these articles will be considered for future studies where we might address long-term effects of ambient air pollution. Those articles are though not related to our investigation of the acute short-term effects of ambient air pollutants in South Africa.

Reviewer 3 Report

While your manuscript has apparently undergone some sort of editorial screening and considered to be of sufficient quality to be sent out for review, I'm still finding many problems with the organization that would lead me to suggest that it still needs editing by someone with some subject area experience.

While the topic is good and the analysis is well-done, the importance of the research needs to be better explained.

I will suggest to the editors that the article be accepted after some major editing that addresses the organization and presentation of the paper.

Author Response

Comments from Reviewer 3

While your manuscript has apparently undergone some sort of editorial screening and considered to be of sufficient quality to be sent out for review, I'm still finding many problems with the organization that would lead me to suggest that it still needs editing by someone with some subject area experience.

While the topic is good and the analysis is well-done, the importance of the research needs to be better explained.

I will suggest to the editors that the article be accepted after some major editing that addresses the organization and presentation of the paper.

Response: Regarding the organization and the explanation of the relevance we adapted the enclosed revision accordingly as those comments are similar to those raised by the first reviewer. The call for editing by someone with “some subject area experience” may need further explanation. Three of the co-authors work since more than 30 years in this field of research and all of them are listed among the worlds “top-100” most cited authors on “[air pollution] and [health]” (web of science). All co-authors contributed to the manuscript.

We look forward to hearing from you in due time regarding our revision and to respond to any further questions and comments you may have.

Round 2

Reviewer 1 Report

I think it has been improved after several revisions.